# Cumulative Incidence of SARS-CoV-2 in Healthcare Workers at a General Hospital in Germany during the Pandemic—A Longitudinal Analysis

**DOI:** 10.3390/ijerph19042429

**Published:** 2022-02-19

**Authors:** Martin Platten, Albert Nienhaus, Claudia Peters, Rita Cranen, Hilmar Wisplinghoff, Jan Felix Kersten, Alexander Daniel Bach, Guido Michels

**Affiliations:** 1Laboratory Dr. Wisplinghoff, 50931 Cologne, Germany; m.platten@wisplinghoff.de (M.P.); h.wisplinghoff@wisplinghoff.de (H.W.); 2Competence Center for Epidemiology and Health Services Research for Healthcare Professionals (CVcare), Institute for Health Services Research in Dermatology and Nursing (IVDP), University Medical Center Hamburg-Eppendorf (UKE), 20246 Hamburg, Germany; c.peters@uke.de (C.P.); j.kersten@uke.de (J.F.K.); 3Department for Occupational Medicine, Hazardous Substances and Health Sciences (AGG), Institution for Statutory Accident Insurance in the Health and Welfare Services (BGW), 22089 Hamburg, Germany; 4Occupational Medicine, St. Antonius Hospital Eschweiler, 52249 Eschweiler, Germany; rita.cranen@sah-eschweiler.de; 5Institute for Virology and Clinical Microbiology, University of Witten/Herdecke, 58448 Witten, Germany; 6Clinic for Plastic and Aesthetic Surgery, Hand and Reconstructive Surgery, St. Antonius Hospital Eschweiler, 52249 Eschweiler, Germany; alexander.bach@sah-eschweiler.de; 7Clinic for Acute and Emergency Medicine, St. Antonius Hospital Eschweiler, 52249 Eschweiler, Germany; guido.michels@sah-eschweiler.de

**Keywords:** occupational health, COVID-19, healthcare worker, vaccination, risk factors, longitudinal study

## Abstract

Health workers (HW) are at increased risk for SARS-CoV-2 infection. In order to monitor the infection dynamic on the basis of contact with patients, HW at the St. Antonius Hospital (SAH) were tested four times in one year by PCR and serology. The cumulative incidence of infection in HW was calculated. Swab and blood tests were simultaneously performed between April 2020 and April 2021. Risk factors and demographic information were assessed at the beginning of the study. The response rate was above 75% in all rounds of testing. The study comprised 1506 HW, 165 (10.6%) of which tested positive for SARS-CoV-2 infection. Working in an ICU or on wards with patient contact were risk factors (OR 4.4, 95% CI 1.73–13.6 and OR 2.9, 95% CI 1.27–8.49). At the end of the study, the majority of HW (810 of 1363 (59.4%)) had been vaccinated at least once. A total of 29.1% of unvaccinated HW and 5.3% of vaccinated HW showed an immune response typical for natural SARS-CoV-2 infection. Of the 73 HW who provided information on the course of the disease, 31.5% reported that their quality of life continued to be impaired. The cumulative incidence of infection was low in these HW, which may be attributed to vaccination and good hygiene. Nevertheless, a work-related infection risk was identified, highlighting the need to improve protection against infection. A high risk of developing long COVID was found after the infection has subsided. Special rehabilitation programs should be provided and HW should be compensated for reduced work capacity in the case that rehabilitation fails or takes a long time.

## 1. Introduction

SARS-CoV-2 continues to sweep through the world and remains a substantial challenge for global health. It is imperative that we bring the pandemic under control and prevent the rise of virus variants that are resistant to the vaccines, which have been developed at an unprecedented pace to fight the virus.

Health workers (HW) have a substantial risk of contracting SARS-CoV-2 due to their frequent and intensive contact with infected patients [1]. In Northern Italy, which was particularly hit by the first COVID-19 wave in Europe, seroprevalence was 28.5% in HW with a high exposure to infected patients compared to 12.8% in the low exposure group [2]. In Spain, similar infection rates were found. In a hospital-based study, 31.6% of the HW were seropositive for SARS-CoV-2, where the odds ratio for infection was highest for doctors (OR = 2.4) [3]. In a French study, the odds ratio for a positive PCR was 3.1 for HW [4]. In a study from the U.K., the relative risk of developing COVID-19 for HW was 7.1 [5]. In a study from Scotland, the hazard ratio for hospitalisation due to COVID-19 was 3.3 for HW [6]. Using insurance data from the U.S., Zhang confirmed an increased risk for HW for having COVID-19 [7]. German health insurance data showed that sick leave or hospitalisation due to COVID-19 was 2.4 times more likely in HW than in other professions [8]. Again, this observation was corroborated by a European study covering nine countries. This compiled data also showed an increased risk for hospitalisation and severe disease for HW [9]. 

According to a systematic review and meta-analysis, infected HW were most often female (78.6%). In addition, the authors summarised studies showing that faulty handwashing, inadequate use of masks and personal protection equipment (PPE), and a shortage of PPE increased the risk of infection for HW, while adequate use of PPE reduced the risk of infection [10]. The various factors that can influence the infection risk of HW in addition to the risk of HW infecting patients, relatives, or each other [4,6,11] make it important to monitor the infection dynamic of HW. A further reason for monitoring the situation closely is the existence of long COVID, also called post-COVID syndrome (PCS) [12]. PCS may influence HWs’ return to work and performance after they recuperate from acute COVID-19, since even in patients with mild or moderately acute COVID-19, fatigue and memory problems are frequent long-term symptoms [13].

In Germany, the seroprevalence rate of HW after the first wave was rather low. In our study, the cumulative incidence of a positive PCR or a positive antibody test was as low as 3.9%. Again, HW with contact to COVID-19 patients had an increased risk of infection [14]. In a study conducted at the beginning of the pandemic from Bonn (study period 9 March–30 April 2020), 1% of HW were infected [15]. Comparatively, the infection rate was 1.8% in Hamburg (study period 20 March–17 July 2020) and 1.6% in Essen (study period 25 March–21 April 2020), whereby in Essen, only staff who had direct contact with SARS-CoV-2 patients were tested [16,17].

In order to monitor the infection dynamic in HW during the first year of the pandemic, we performed a longitudinal analysis at a large hospital. In Germany, as in many other countries, HW are a high-priority group for vaccination [18]. In addition, willingness to be vaccinated against COVID-19 is high among German HW [19]. Therefore, we started a vaccination campaign at the hospital as soon as the vaccine became available in Germany at the beginning of 2021. In addition to the dynamics of the SARS-CoV-2 infection in HW during the pandemic, we also analysed the vaccination’s uptake, its side effects, and vaccine-induced antibodies.

## 2. Materials and Methods

The study took place at St. Antonius Hospital (SAH) from April 2020 to April 2021. The SAH is a general hospital with 1363 staff and 443 hospital beds. The SAH offers a wide range of diagnostics and treatments at the university level to the population in the surrounding county. The SAH is located near Aachen, a city close to the border of the Netherlands, where on 15 February 2020, an indoor celebration with 300 participants as part of the carnival festivities took place, which was assumed to be responsible for the first large COVID-19 outbreak in Germany. As the population concerned is served by the SAH, it seemed interesting to investigate the infection dynamic of the staff of the SAH. Aside from this scientific interest, the staff themselves also expressed an interest in being monitored for SARS-CoV-2 infections.

All 1363 staff of the SAH were invited to participate in this follow-up study, regardless of patient contact. The interviews and tests were organised by the occupational medicine consultant of the SAH (RC) and her team. They were supported by the head of the emergency department (GM) and team. Participants were informed in advance about study goals and procedures as well as data protection. There were no exclusion criteria. Participation was voluntary, and there were no financial or other incentives for participation. All participants gave their written consent after receiving detailed verbal and written information. The study was funded by the Institution for Statutory Accident Insurance and Prevention in the Health and Welfare Services, Hamburg, Germany (Berufsgenossenschaft für Gesundheitsdienst und Wohlfahrtspflege (BGW) project number: ext. FF_1461). The ethics committee of the medical association in Hamburg was consulted, and there were no concerns regarding the study conduct. The study was conducted between 27 April 2020 and 23 April 2021, with four instances (rounds) of sample collection.

### 2.1. Study Design

In the first round (April/May 2020), the participants completed a questionnaire developed by the study team which asked for their socio-demographic data, area of operation, pre-existing conditions, contact with SARS-CoV-2 positive patients or co-workers, and SARS-CoV-2-specific symptoms. In the last round, 73 participants who had had a confirmed past SARS-CoV-2 infection completed a short questionnaire on symptoms that are characteristic of PCS. The questionnaire used dichotomous questions to ask the participants about their health at the time of the fourth round, including their quality of life, general health, physical fitness, tiredness, memory problems and shortness of breath following infection.

When we started vaccinating in January 2021, the participants were asked to fill out a short questionnaire about the side effects of the vaccine. The possible answers were that they had experienced no side effects or had experienced local, systemic or local and systemic side effects. The type of vaccine was documented and the number of days of sick leave following the vaccination was assessed. 

We collected a deep nasopharyngeal swab and a serum sample at each of the four sampling rounds (April/May, September and November/December 2020, and April 2021). Nasopharyngeal swabs were tested for SARS-CoV-2 RNA by reverse transcription polymerase chain reaction (RT-PCR) [20]. SARS-CoV-2 serology was performed on participants at all four study visits. Serum samples were tested qualitatively for SARS-CoV-2 nucleocapsid antibodies (IgG and IgA) in three study visits (April to December 2020) using a commercially available enzyme-linked immunosorbent assay (ELISA) [21]. In the context of the last study visit (April 2021), with the advent of new assays, we were able to quantitatively assess IgG antibodies against the SARS-CoV-2 spike protein [22]. In April 2021, all participants with a positive SARS-CoV-2 IgG serology were tested for IgM/IgG antibodies against SARS-CoV-2 nucleocapsid [23] in order to differentiate between serological response to vaccination and natural infection. Only participants with a positive serology against nucleocapsid were classified as having been through a natural infection. A constellation of positive serology against the spike protein without detection of nucleocapsid antibodies was interpreted as a response to vaccination. Participants with a positive PCR were classified as having acute SARS-CoV-2 infection. Individuals with isolated detection of IgA during the first three study visits were not deemed SARS-CoV-2-positive, as IgA has low specificity [24]. All assays were conducted in compliance with the manufacturer’s specifications.

### 2.2. Data Management and Statistical Methods

Collected data were pseudonymised using a digital code. The association of data with participants was only possible via an encrypted identification list that was deposited electronically with the supervising Occupational Health Consultant (RC). After study completion, all data shall be stored for ten years according to regulations. Serum samples will be destroyed after a maximum of two years. We adhered to all data protection regulations, guaranteed doctor−patient confidentiality, and followed the declaration of Helsinki. 

Missing values were indicated in the tables. Metric variables were described in terms of arithmetic mean, median, standard deviation (SD), and range. Categorical variables were described with absolute and relative frequencies. Group differences were determined by chi-squared test, and in the case of a sparse cell composition, by Fisher´s exact test. Logistic regression was used for multivariate analysis of binary outcomes, and odds ratios (OR) with corresponding 95% confidence intervals (95% CI) were calculated. Modelling was adjusted for age and gender. A *p*-value ≤ 0.05 was deemed statistically significant. Data were analysed using SPSS software (version 27, SPSS Inc., Armonk, NY, USA) for the descriptive part of the analysis and the logistic regression. R (version 4.1.1., R Foundation for Statistical Computing, Vienna, Austria) was used [25] for visualising the results.

## 3. Results

The study initially comprised 1212 of 1363 (88.9%) eligible staff (Figure 1). Of note is the fact that there was some turnover of staff during the study period. This explains the fluctuation of the number of study participants over the study period. 1506 HW participated in at least one test; 649 HW (43.1%) participated in all four tests, and 287 (19.1%) participated in one of the four rounds of testing (Table 1). The majority of participants were female (52.6%), and gender was unknown for 33.6% of participants. The mean age was 43.7 years, with a range between 18 and 75. A total of 125 HW (8.3%) worked in intensive care units (ICU). The workplace was unknown for 40.6% of participants. Moreover, 113 (7.5%) HW worked in departments with patient contact. 

Before the study started, 16 staff members (1.2%) had tested positive for SARS-CoV-2 RNA (Table 2). Cumulatively, a total of 52 of 1506 (3.5%) HW had tested positive for SARS-CoV-2 RNA by April 2021. At the different study visits, SARS-CoV-2 RNA was detected in 3 (0.2%), 8 (0.8%), 21 (2.0%), and 4 (0.4%) participants, respectively.

In the first three study visits, IgG was detected in 40 (3.3%), 31 (31/1023, 3.0%), and 56 (56/1074, 5.2%), and IgA was detected in 105 (8.7%), 61 (6.0%), and 74 (6.9%), respectively. Both IgG and IgA were detected in 31 (2.6%), 15 (1.5%) and 43 (4.0%), respectively (Table 2). In the last test round, IgG against SARS-CoV-2 spike protein was detected in 809 (77.8%). In 103 HW (9.9% of all participants), antibody testing against SARS-CoV-2 nucleocapsid indicated a natural infection. In 706 HW (67.9%), IgG was positive because of vaccination.

The cumulative incidence in our cohort was 11.0%. Out of 165 HW concerned, information about their workplace was available for 120 (72.7%). Regardless of infection status, this information was available for 59.4% of all participants. The cumulative infection incidence was 19.2% for HW in the ICU and 4.4% for staff without patient contact. The OR for infection was 4.42 for working in ICU and was statistically significant. Working on wards with patient contact was also a risk factor (OR 2.92) (Table 3). 

We do not know how many HW of the 165 HW positive in PCR or IgG developed symptoms of acute COVID-19. However, 73 HW (44.2%) answered the questionnaire concerning persisting symptoms (Table 4) and indicated weariness or tiredness (47.9%) most frequently, followed by memory problems (42.5%). Physical fitness was still reduced in 35.6% and quality of life was diminished in 31.5%, while 30.1% reported shortness of breath.

By April 2021, 810 (77.8%) of the HW at SAH taking part in the fourth round had been vaccinated with one dose of COVID-19 vaccine and 397 (38.2%) with two doses (Table 5). For those vaccinated with the mRNA vaccine by BioNTech/Pfizer, local reactions were the most frequently mentioned side effects after the first vaccine dose (53.9%), and local and systemic reactions were most frequently mentioned (31.3%) after the second vaccine dose. After the first dose of the vector vaccine by AstraZeneca, 50.3% of the HW reported local and systematic reactions. Seven HW received the vector vaccine for the second dose, of which four reported no side effects, and data are missing for three HW.

From a total of 1207 vaccinations (first and second vaccine doses), 145 HW reported taking sick leave after vaccination (12.0%). The highest rate of sick leave was observed after the first mRNA dose (14.7%). For the second vaccine dose with mRNA, the rate was 11.8%. For the vector vaccine, the sick leave rate was 10.7% (Table 6). Most of the time, the sick leave ended after one or two days. A sick leave of three to seven days was rare (0.8% to 2.5%, depending on the vaccine and number of doses).

In the unvaccinated group, 60 of 206 HW (29.1%) had an increased IgG concentration (Figure 2). Following the first vaccine dose, 330 of 407 HW (81.1%) had an increased IgG concentration; following the second vaccine dose, the figure was 397 of 398 HW (99.7%).

## 4. Discussion

This is the first longitudinal study to collect comprehensive data on SARS-CoV-2 infection developments using PCR and serology in an occupational context among staff in a German hospital. SARS-CoV-2 infection was evident in 165 (11.0%) study participants. The incidence peaked during the third test round in November/December 2020, matching the time of the second wave of COVID-19 in the German population. Working in an ICU and working on wards with patient contact were risk factors for SARS-CoV-2 infection (OR 4.4, 95% CI 1.73–13.6 and OR 2.9, 95% CI 1.27–8.49). At the end of the study, the majority of HW (810 of 1363 (59.4%) had been vaccinated at least once. At this time, 26.1% of the unvaccinated HW were serologically positive compared with 5.3% of the vaccinated HW who showed an immune response typical of natural SARS-CoV-2 infection.

### 4.1. PCR

The yield of positive PCR results was rather low. In total, we tested 1506 HW at least once and found 52 (3.5%) positive results via PCR. Of these 52 positive HW, 16 had already been identified because of a symptom-related screening prior to the start of the study. This further reduced the yield of the repeated mass screening during the study (36 positive PCR or 2.4% of all participants). Considering the number of PCR tests performed during the study (*n* = 4348), the rate of a positive PCR was as low as 0.4%. However, it is worth noting that the highest number of positive PCR in HW was found during the second wave of COVID-19 in the German population in November and December 2020 (*n* = 21, 2%). Similar results were reported in the U.K. When screening asymptomatic HW, the highest rate of positive PCR was found when the infection rate was the highest in the general population [26]. The rate of positive PCR in this study varied between 1.1% and 7.1%. Considering these values, it seems likely that a point of care (PoC) test followed by a PCR is more efficient for screening asymptomatic HW [27]. How the Omicron variant might change this perspective remains to be verified. It seems screening HW with symptoms increases the yield of positive PCR. Again, in a U.K. study, up to 20% of HW with typical symptoms tested positive with PCR [28]. 

During the last round of tests in March and April 2021, Germany was in the third wave of COVID-19. However, the number of positive PCRs for the HW in our study was lower than during the previous round (*n* = 4, 0.4% compared to *n* = 21, 2%). At this time, 38.2% of the HW had been vaccinated twice and 39.6% once. The first vaccine dose already reduces the infection risk in HW, as has been demonstrated, for example, in data from the U.S. [29]. This might have prevented some infections of HW at SAH during the third wave, which was even more severe than the two previous waves in the German population. This is supported by the observation that the rate of positive PCR in March/April 2021 was higher in unvaccinated HW than in vaccinated HW (0.9% versus 0.2%). However, the case number was too small to draw strong inferences.

### 4.2. Seroprevalence

During the course of the study, the prevalence of IgG increased from 3.3% to 9.9%. The low IgG prevalence in HW in Germany after the first COVID-19 wave has been confirmed by other studies [15,16,17]. In a six-month follow-up study during the summer of 2020—a period of low COVID-19 incidence in the general population of Germany—the rate of positive IgG in HW was 0.75% [30]. A similar study performed in another region of Germany that covered the period of six months from July to December 2020, thereby including the second wave of COVID-19, reported a seroprevalence of 13.4% [31]. The prevalence rate was significantly higher among HW working in worksites with more frequent contact with confirmed or suspected cases (30.3%, *p* = 0.003). A longitudinal study from Hamburg, Germany, which reported seroconversion rates observed in May 2021, found a seroprevalence of 4.7% [16]. The low rate of infection in HW at the SAH might be explained by several factors. At the SAH, hygiene rules were swiftly established and stringently monitored. Special emphasis was placed on repeated consensus-based training by hospital hygiene staff on basic hygiene and the handling of personal protective equipment. Communication was also an important aspect. Weekly online training and briefing of all staff via newsletters (regarding the development of international and national case numbers, current studies, current recommendations from the Robert Koch Institute, regional recommendations, and in-house decisions by the crisis team) might have increased the awareness and knowledge of preventive behaviour among HW.

Though the cumulative incidence of SARS-CoV-2 was low in our study, and despite all the training efforts, HW with regular patient contact or who worked in an ICU had an increased infection risk. This is consistent with results from other studies which observed increased infection risks in HW [1,2,3,4,5,6,7,8,9,31]. Therefore, studies on continuous improvement of infection prevention and control for HW are needed. Moreover, it also seems important to provide compensation for HW with health problems after the SARS-CoV-2 infection. As in most countries, these will be judicial decisions; our data indicate that HW with close patient contact should be considered as having an increased infection risk when assessing such claims. 

### 4.3. Long COVID or Post-COVID Syndrome (PCS)

Our data indicate that PCS might be a problem in HW. Out of those who completed the questionnaire (44.8% of all potentially eligible HW), weariness or tiredness (47.9%) was most frequently reported, followed by memory problems (42.5%) and shortness of breath (30.1%). A total of 35.6% of participants described their physical fitness as still reduced, and 31.5% said their quality of life had diminished. Similar results are reported in a study from the UK. Out of 3759 HW tested, 932 (24%) were infected with SARS-CoV-2. After three to four months, 39% still reported severe fatigue and 40% reported mild to moderate shortness of breath [32].

Tiredness or fatigue is one of the most frequently described long COVID symptoms in the literature [33]. PCS was reported more frequently in COVID-19 patients who were hospitalised than those not hospitalised. However, PCS can also have a negative influence on the social and occupational life of HW with a mild course of acute COVID-19, as has been shown in a Swedish study [13]. Our findings should be verified in additional studies focusing on PCS in HW. In particular, attention should be paid to the financial consequences of COVID-19 and PCS. As is pointed out by a working group from India, one-third of the HW are worried about the financial impact of PCS on them and their families [33]. 

### 4.4. Vaccination

Acceptance of vaccination by the HW at SAH was high. Three months after the start of the vaccination campaign, 77.8% of the HW had been vaccinated at least once. This high willingness of HW to be vaccinated has been confirmed in other studies [19,34]. However, an early survey started before the vaccines became available found that only 65% were inclined to be vaccinated [35]. Again, the SAH’s good communication strategy might have helped to create a positive attitude towards vaccination. 

There were side effects from vaccination in three out of four HW who returned the questionnaire. However, we observed no severe side effects, and most side effects were local and of short duration. The number of side effects did not increase with the second vaccine dose. It should be noted that up to 14% of HW took sick leave following vaccination, which mostly lasted no longer than two days. Sick leaves between three and seven days were rare (0.8% to 2.5%), and none lasted longer than seven days. Our data are consistent with the findings of an Italian study, which also observed few HW requests for sick leave following vaccination. The authors reported a rate of 1.6% after the first vaccine dose and 6.1% after the second [36]. Sick leave following vaccination is certainly an important aspect to consider, but it should be regarded in light of the fact that vaccination reduces HW sick leave due to infection and that it most likely reduces the duration of sick leave when infection occurs despite vaccination [37].

After the first vaccine dose, 81.1% of HW had a significant immune response, possibly indicating that there was already some degree of protection [29]. After the second vaccine dose, only one serological non-responder was found (0.3%). In the literature, non-responders are described in patients with immune deficiencies [38,39,40]. We found no study that investigated non-responders in HW. Based on our data, this topic is of minor concern for HW.

The data from our study cover the period of the pandemic from its beginning until April 2021. In the beginning, little was known about the best strategy to monitor HW for SARS-CoV-2 infection. PCR testing has the merit of being able to identify asymptomatic but contagious persons. However, to monitor the infection dynamics, PCR tests have to be repeated quite often, which is challenging for the HW, the organisation, and the budget. The yield of cases in PCR was rather low in our study, indicating that regular PCR testing might not be cost-effective. This is also pointed out by a recent cost-effectiveness study concerning SARS-CoV-2 screening. The authors concluded that IgG-based screening combined with symptom-based PCR testing is more efficient than PCR screening alone [41]. In our study, we identified 154 HW with SARS-CoV-2 infection using IgG blood tests. Out of these 154 HW, 113 (73.4%) were positive in IgG but not positive in the PCR test, which is in line with the findings of the cost-effective study mentioned above. This might explain why only a few studies could be found that tested HW using both PCR and IgG. A study from India tested 190 samples from HW for RT-PCR using nasopharyngeal swabs, and all 190 samples tested negative for RT-PCR. These samples were also screened for SARS-CoV-2-IgG antibodies, and 48 (25.3%) were found to be reactive [42]. In a register-based study from Denmark, HW were screened from April to October 2020, targeting SARS-CoV-2 total antibodies. Data on all participants’ PCR tests for SARS-CoV-2 RNA were obtained from national registries. With few exceptions (2.4%), all HW with a positive PCR test developed antibodies [43]. As with the study from India, the design of the Danish study was different to the design of our study.

In Germany, the incidence of COVID-19 surged again in November and December 2021. The new wave was dominated by the Delta variant of SARS-CoV-2, which is more infectious and causes more severe symptoms of COVID-19 than the previous variants. This also affected HW, as was shown in a study from the U.S. [44]. In Germany, this fourth wave turned into a fifth wave, being dominated by Omicron. This time, the virus variant is even more infectious than Delta but seems to cause fewer severe reactions to the disease. In addition, vaccination against COVID-19 seems to protect against Omicron [45]. However, data from Israel show that due to a combination of new, more aggressive variants of the virus and dilution of the vaccine’s effectiveness over time regardless of the vaccination regime applied, a booster is necessary [46,47]. How these developments influence infection prevention in HW needs to be monitored in future. Our data show that vaccination and testing are important tools for the protection of HW. Nevertheless, they will not work without other vigorous organisational, technical, and personal protective measures. As we also observed in infections in HW using PPE, it seems important to regularly train HW in the proper use of PPE [14].

### 4.5. Limitations and Strengths

This study was conducted in real-world settings. Testing, interviews, and data collection had to be organised in such a way as to avoid further complicating the already strained routine in the hospital by the study. Therefore, a risk assessment was only performed during the first round. This is a limitation, as we had to restrict the calculation of risk factors for infection to a subgroup. In addition, the exposure situation may have changed over time, which could have introduced non-differential misclassification and, therefore, a potential for underestimation of risks, but we were not able to assess this. After HW were affected with COVID-19, the occupational medicine department closely consulted with them. However, their long-term symptoms were only assessed by means of a short, standardised questionnaire, with responses from 44.8% of the potentially eligible HW. Furthermore, the reasons for sick leave following vaccination were not assessed: this should be kept in mind when drawing inferences concerning the side effects of the COVID-19 vaccination.

The participation rate in all four rounds combined was 43.1%. On the other hand, the response rate in the single rounds was above 76% if the total workforce at the start of the study was taken as a reference; therefore, the potential for selection bias due to participant self-selection is likely to be limited. The relatively high response rate made it possible to describe the occurrence of infection in the workforce at large.

## 5. Conclusions

The infection rate in HW at the SAH was relatively low even before the COVID-19 vaccines became available. Nevertheless, working in the ICU or regular patient contacts were risk factors for infection in HW. Therefore, infection prevention needs to be improved. Although the vaccines were well tolerated, some HW still took sick leave after the vaccinations. It was our intention to complete our longitudinal study upon completion of the vaccination campaign. In the meantime, however, it has become apparent that a third vaccine dose is now needed. According to our current knowledge, the infection protection by the vaccine is dependent on the virus variant (Delta or Omicron) and the time elapsed following vaccination. It is therefore necessary to continue monitoring the infection risk in HW. Though the number of HW with COVID-19 in our study was small, the data indicate that the health and quality of life of some affected HW are impeded in the long term. Support, rehabilitation, and, if needed, financial compensation for HW with PCS need to be provided.

## Figures and Tables

**Figure 1 ijerph-19-02429-f001:**
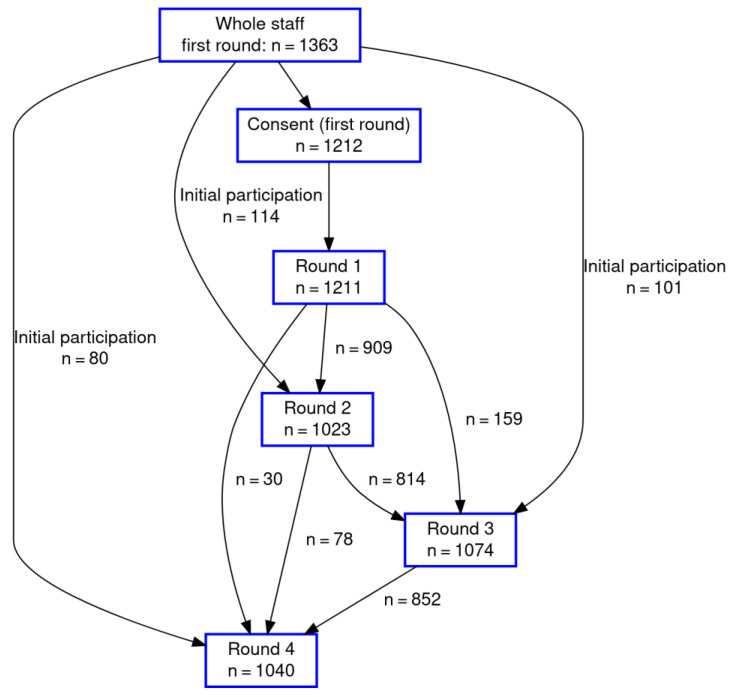
Flow chart of 1506 participants of the prospective study.

**Figure 2 ijerph-19-02429-f002:**
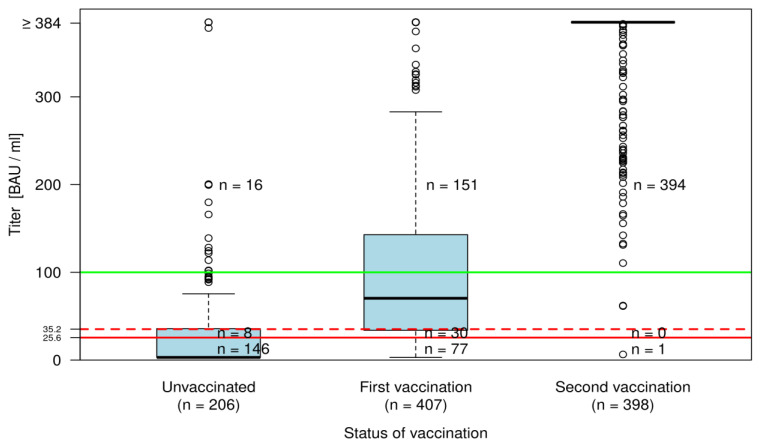
IgG titres in unvaccinated and vaccinated HW with first and second vaccination dose. (*n* = number of cases falling in the particular categories: 0–<25.6, 25.6–<35.2, 35.2+ BAU/mL by status of vaccination).

**Table 1 ijerph-19-02429-t001:** Description of the study population.

Number of Rounds Participated	*N*/Mean	%/Range
1	287	19.1
2	246	16.3
3	324	21.5
4	649	43.1
Gender		
Male	206	13.7
Female	792	52.6
Unknown	508	33.6
Age (mean ± SD) [min.–max.]	43.7 ± 13.2	18–75
Ward		
Intensive care (with patients requiring ventilation)	125	8.3
Wards or tasks with patient contact	657	43.6
Tasks without patient contact	113	7.5
Unknown	611	40.6
Total	1506	100.0

**Table 2 ijerph-19-02429-t002:** Results of PCR and IgG or IgM in vaccinated participants, fourth round.

	PCR+		IgG+		IgG or PCR+		Participants
Time of Test	*N*	%	*N*	%	*N*	%	*N* (%)
Round 1	3 (16) *	1.6	40	3.3	47	3.9	1211
Round 2	8	0.8	31	3.0	38	3.7	1023
Round 3	21	2.0	56	5.2	62	5.8	1074
Round 4 (IgM+) *							
All	4	0.4	103	9.9	103	9.9	1040
Not vaccinated (IgG)	2	0.9	60	26.1	60	26.1	230 (22.1)
Vaccinated (IgG/IgM) **	2	0.2	43	5.3	43	5.3	810 (77.9)
Either PCR or Ig positive	52	3.5	154	10.2	165	11.0	1506

* 16 HW were positive in PCR before the study started. ** a total of 809 (77.8%) HW tested positive for the spike protein at round 4; 706 (67.9%) HW tested positive due to vaccination.

**Table 3 ijerph-19-02429-t003:** Logistic regression with ward as risk factor for SARS-CoV-2 infection (*n* = 895, IgG or PCR+ *n* = 120).

	IgG and PCR−		IgG or PCR+		Logistic Regression *	
Ward or Task	*N*	%	*N*	%	OR	95% CI
Intensive care unit (ICU)	101	80.8	24	19.2	4.42	1.73–13.6
Normal care	566	86.1	91	13.9	2.92	1.27–8.49
No patient contact	108	95.6	5	4.4	1	--

* adjusted for age and gender.

**Table 4 ijerph-19-02429-t004:** Health and symptoms at the final visit of HW with COVID-19 (*n* = 73).

Symptom	*N*	%
Quality of life still diminished	23	31.5
General health still diminished	21	28.8
Physical fitness reduced	26	35.6
Weariness, tiredness increased	35	47.9
Memory problems increased	31	42.5
Shortness of breath increased	22	30.1

**Table 5 ijerph-19-02429-t005:** Number of vaccinations by vaccine.

	BioNTech/Pfizer	AstraZeneca	Other	All
	*N* (%)	*N* (%)	*N* (%)	*N* (% *)
First vaccine dose	397 (49.0)	401 (49.5)	12 (1.5)	810 (77.8)
Second vaccine dose	382 (96.2)	7 (1.8)	8 (2.0)	397 (38.2)

* % of 1040 participants at the fourth round.

**Table 6 ijerph-19-02429-t006:** Side effects of vaccination by vaccine and number of doses.

	BioNTech/Pfizer		AstraZeneca	
First Vaccination	*N*	%	*N*	%
No	95	24.1	45	11.5
Local	213	53.9	41	10.5
Systemic	22	5.6	109	27.8
Local and systemic	65	16.5	197	50.3
Total *	395	100.0	392	100.0
Sick leave in days				
1	28	7.1	23	5.9
2	20	5.1	12	3.1
3–7	10	2.5	4	1.0
Sick leave, all	58	14.7	42	10.7
Second vaccination				
No	75	19.7	4	100.0
Local	100	26.3	0	0.0
Systemic	86	22.6	0	0.0
Local and systemic	119	31.3	0	0.0
Total **	380	100.0	4	100.0
Sick leave in days				
1	23	6.1	0	--
2	19	5.0	0	--
3–5	3	0.8	0	--
Sick leave, all	45	11.8	0	--

* 9 missing values (2 for mRNA, 7 for vector). ** 5 missing values (2 for mRNA, 3 for vector). In total, there were 16 missing values out of 1207 possible responses (1.6%), and 145 HW reported sick leave after vaccination (12.0%).

## Data Availability

The data are available upon request from the corresponding author.

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
