# Peer review of "Cumulative Incidence of SARS-CoV-2 in Healthcare Workers at a General Hospital in Germany during the Pandemic—A Longitudinal Analysis"

_ijerph, 2022, doi:10.3390/ijerph19042429_

Round 1
Reviewer 1 Report
- At the end of the abstract, please discuss your finding of the research. Please suggest to the policymaker.
- In the introduction section, please discuss the more recent issue related to the corvid-19, in 2011.
- what do you mean "All employees of all departments of the SAH were invited to participate in a longitu- 91 dinal investigation within the context of occupational healthcare" not clear this line. Need to rewrite again.
- There is required more critical recent literature and based on theoretical argumentation.
- The author (s) should discuss the limitations of this study and future research direction in a constructive way.
- The author (s) did not discuss the theoretical and practical contribution of this study. Author(s) should discuss this study's theoretical and practical contribution in the separate subsection under discussion for more clarity.
- The author should provide a precise conclusion section.
Author Response
Reviewer 1
1 At the end of the abstract, please discuss your finding of the research. Please suggest to the policymaker.
Answer:
We agree, the conclusion was kind of evasive. Now we state the conclusion more vigorous.
The cumulative incidence of infection was low in these HW. Vaccination and good hygiene are probable explanations. Nevertheless, a work related infection risk was detected. This indicates that infection protection need to be further improved. The rate of infected HW with Long COVID was high. Special rehabilitation programs need to be provided and HW should be compensated for reduced workability if rehabilitation fails or takes a long time.
2 In the introduction section, please discuss the more recent issue related to the covid-19, in 2011.
Answer:
Thank you for the suggestion. Now we discuss newly emerged questions, i.e. Delta and Omega, duration of vaccination effectivity and booster in an added paragraph in the discussion. As we present data from the first year of the pandemic, it is suitable to discuss our findings in the light of newly emerged or looming challenges. See added paragraph before the conclusion.
3 what do you mean "All employees of all departments of the SAH were invited to participate in a longitudinal investigation within the context of occupational healthcare" not clear this line. Need to rewrite again.
Answer:
Thank you for the comment. We rephrased the sentence.
All 1,363 employees of the SAH were invited to participate in this follow-up study, regardless of having patient contact. The interviews and tests were organised by the occupational medicine consultant of the SAH (RC) and her team. They were supported by the head of the emergency department (GM).
4 There is required more critical recent literature and based on theoretical argumentation.
Answer: We added recent literature on the virus variants Delta and Omega, vaccination and test strategy. (see answer to suggestion 2.)
5 The author (s) should discuss the limitations of this study and future research direction in a constructive way.
Answer:
Limitations we discuss in chapter 4.5.
Now we added to this chapter:
Therefore, risk assessment was only performed during the first round. This is a limitation, as we had to restrict calculation of risk factors for infection to a subgroup. In addition, the exposure situation might have changed over time. We were not able to assess that. This has most likely introduced non-differential misclassification and therefor potential for underestimation of risks.
Further research direction is discussed in the last two paragraph of the discussion now.
6 The author (s) did not discuss the theoretical and practical contribution of this study. Author(s) should discuss this study's theoretical and practical contribution in the separate subsection under discussion for more clarity.
Answer: Thank you for this suggestion. The study’s contributions are now mentioned in the separate subsections of the discussion and the last two paragraphs, newly added, of the discussion.
7 The author should provide a precise conclusion section.
Answer:
Thank you for your comment. We tried to be more precise with our conclusion.
We added that improved infection prevention is needed and that rehabilitation and support is needed for HW with Long-COVID.
Reviewer 2 Report
The report „Cumulative Incidence of SARS-CoV-2 in Healthcare Workers at a General Hospital in Germany During the Pandemic – A Longitudinal Analysis” give us an overview on cumulative incidence of SARS-CoV-2 in healthcare workres in german hospital in Aachen. Among others it compares side effects of vaccination by BioNTech/ Pfizer nad Astra Zeneca vaccine and shows the frequency of acute COVID-19 symptoms. After a few corrections I recommed to publish this work.
The percentage of unvaccinated HW is different in the text and table 2. Which one is correct?
Table 5 and 6 could be merged. There is no need to produce so many tables.
Line 192: shortness breath (30,1) should be indicated after physical fitness (35.6%).
In conclusions section explain better why a third vaccination is needed.
Author Response
Reviewer 2
1 The report „Cumulative Incidence of SARS-CoV-2 in Healthcare Workers at a General Hospital in Germany During the Pandemic – A Longitudinal Analysis” give us an overview on cumulative incidence of SARS-CoV-2 in healthcare workres in german hospital in Aachen. Among others it compares side effects of vaccination by BioNTech/ Pfizer nad Astra Zeneca vaccine and shows the frequency of acute COVID-19 symptoms. After a few corrections I recommed to publish this work.
Answer:
Thank you for this positive feedback.
2 The percentage of unvaccinated HW is different in the text and table 2. Which one is correct?
Answer:
Thank you for pointing this out. The number in the table is correct. We changed the number in the text from 77.9 to 77.8.
3 Table 5 and 6 could be merged. There is no need to produce so many tables.
Answer:
We prefer not to merge table 5 and 6. Table 5 has two more columns than table 6.
4 Line 192: shortness breath (30,1) should be indicated after physical fitness (35.6%).
Answer:
Thank you for pointing this out. It was changed accordingly
5 In conclusions section explain better why a third vaccination is needed.
Answer: Thank you for the suggestion. At the end of the discussion, we now explain why a booster is needed. This supports our conclusion
Reviewer 3 Report
The article discusses a very important issue such as disease incidence of healthcare workers on covid.
The strength of the work is the analysis of the literature, which is very fresh and interesting. The weakness is the presentation of the methodology and especially of the respondents. I propose to slightly characterize the location of the hospital and its size, where the study is carried out.
Authors write that: "All employees of all departments of the SAH were invited to participate in a longitudinal investigation within the context of occupational healthcare" - I propose to put the number of employees.
The same situation here. Authors write that: "In the last round, participants who had had a proven SARS-CoV-2 infection" - lack information how many participant had a proven of sars.
In the descriptive part of the research, the respondents are not characterized in any way. It is sorely lacking. I propose to complete the respondents in terms of gender, age, seniority, incidence in terms of covid and hospital hours. I think that these factors are the added value of the research carried out and it is worth describing them.
I also propose to describe what data analysis using SPSS software is all about.
In discussion very interested is: "This is the first longitudinal study to collect comprehensive data on SARS-CoV-2 infection developments using PCR and serology in an occupational context among workers in a German hospital." I propose to refer to studies in other countries at this point. It is interesting that this type of research is the first in Germany, so it is worth noting whether such research was carried out in other countries and what is the result of it.
I also propose to attach a survey to the work, which was carried out as an attachment, which will also be the value of the work.
Author Response
Reviewer 3
1 The article discusses a very important issue such as disease incidence of healthcare workers on covid.
Answer:
Thank you for this positive remark.
2 The strength of the work is the analysis of the literature, which is very fresh and interesting. The weakness is the presentation of the methodology and especially of the respondents. I propose to slightly characterize the location of the hospital and its size, where the study is carried out.
Answer:
Thank you for this remark. We added the information. Now we write: The SAH is a general hospital with 1,363 employees and 443 hospital beds. The SAH offers a wide range of diagnostics and treatments at university level to the population in the surrounding county. The SAH is located near Aachen, North Rhine-Westphalia, Germany. Aachen is the hometown of Charlemagne, crowned by the pope in 800 AD. Aachen is located close to the boarder of the Netherlands. In this part of Germany, carnival is celebrated intensively with festivities in- and outdoor from November to February each year. On February 15th 2020, one of these indoor festivities with 300 participants took place, which is assumed to be responsible for the first large COVID-19 outbreak in Germany. As the concerned population is served by the SAH, it seems interesting to investigate the infection dynamic in the employees of the SAH. Beside of this scientific interest, the employees themselves expressed an interest in being monitored for SARS-CoV-2 infections. .
3 Authors write that: "All employees of all departments of the SAH were invited to participate in a longitudinal investigation within the context of occupational healthcare" - I propose to put the number of employees.
Answer: We made changes accordingly
4 The same situation here. Authors write that: "In the last round, participants who had had a proven SARS-CoV-2 infection" - lack information how many participant had a proven of sars.
Answer: Now we write: In the last round, 73 participants who …..
5 In the descriptive part of the research, the respondents are not characterized in any way. It is sorely lacking. I propose to complete the respondents in terms of gender, age, seniority, incidence in terms of covid and hospital hours. I think that these factors are the added value of the research carried out and it is worth describing them.
Answer:
Thank you for the remark. In table 1 we describe the demographic characteristics of the cohort. Information on seniority, and hospital hours (working hours?) are not available. Incidence of Covis is described in tables 2 and 3.
6 I also propose to describe what data analysis using SPSS software is all about.
Answer:
Thank you for this remark. We moved the sentence concerning the software used to the end of the paragraph and rephrased it. Now we write: Data was analysed using SPSS software (version 27, SPSS Inc.) for the descriptive part of the analysis and the logistic regression. For visualizing the results R Version 4.1.1. was used [25].
7 In discussion very interested is: "This is the first longitudinal study to collect comprehensive data on SARS-CoV-2 infection developments using PCR and serology in an occupational context among workers in a German hospital." I propose to refer to studies in other countries at this point. It is interesting that this type of research is the first in Germany, so it is worth noting whether such research was carried out in other countries and what is the result of it.
Answer:
Thank you for this comment. So far, only few studies used PCR and IgG in HW simultaneously and none was similar to our study in design. We introduced this at the end of the discussion.
8 I also propose to attach a survey to the work, which was carried out as an attachment, which will also be the value of the work.
Answer:
I am not sure what you mean by “survey of the work”. The questionnaire we used was straight forward and developed for the study. We do not think that it is of value to translate the questionnaire for the supplement. The content of the questionnaire is described in the manuscript.